# The Role of the Ectopeptidase APN/CD13 in Cancer

**DOI:** 10.3390/biomedicines11030724

**Published:** 2023-02-28

**Authors:** Uwe Lendeckel, Farzaneh Karimi, Ruba Al Abdulla, Carmen Wolke

**Affiliations:** Institute of Medical Biochemistry and Molecular Biology, University Medicine Greifswald, University of Greifswald, D-17475 Greifswald, Germany

**Keywords:** aminopeptidase N, angiogenesis, ANPEP, cancer, cancer stem cells, drug resistance, invasion, migration

## Abstract

APN/CD13 is expressed in a variety of cells/tissues and is therefore associated with diverse physiological functions, including proliferation, differentiation, migration, angiogenesis, invasion, metastasis, vasoconstriction, and the regulation of normal and impaired immune function. Increased expression or activity of APN/CD13 has been described for various tumors, such that APN/CD13 is in most cases associated with reduced disease-free and overall survival. The mechanisms that mediate these cellular effects of APN/CD13 have been largely determined and are described here. APN/CD13-regulated signaling pathways include integrin recycling, the regulation of small GTPase activities, cell–ECM interactions, and Erk1/2, PI3K, and Wnt signaling. APN/CD13 is a neo-angiogenesis marker that is not found on normal endothelia, but it is found on neo-angiogenetically active endothelia. Therefore, APN/CD13 represents a specific receptor for so-called “tumor-homing peptides” (NRG peptides). Peptides containing the NRG motif show high-affinity binding to APN/CD13. APN/CD13 thus represents a versatile target for the inhibition of tumor-induced angiogenesis through the tumor-selective administration of, e.g., cytotoxic substances. Furthermore, it enables the molecular imaging of tumor masses and the assessment of (neo)angiogenesis in animal models and in patients. Pharmacological inhibitors of APN/CD13 have been proven to reduce tumor growth and tumor progression in various APN/CD13-positive tumors.

## 1. Introduction

Aminopeptidase N (membrane alanyl aminopeptidase, APN, EC 3.4.11.2, CD13, and ANPEP) is a Zn^2+^-dependent, membrane-bound ectopeptidase that preferentially cleaves small neutral amino acids from the N-terminus of small peptides [1,2]. APN/CD13 is expressed on a variety of cells/tissues and is therefore associated with diverse physiological functions including proliferation, differentiation, migration, angiogenesis, invasion, vasoconstriction, and the regulation of normal and impaired immune function [3]. Increased expression or activity of APN/CD13 has been described for many different tumors [4,5,6], and this has been shown in most cases to be associated with reduced disease-free and overall survival [7,8,9]. However, higher levels of plasma APN activities are associated with better survival in colorectal cancer.

APN/CD13 is a neo-angiogenesis marker not found in normal vasculature, but found in newly formed vasculature (angiogenetically active endothelium). APN/CD13 is therefore an interesting and specific receptor for so-called “tumor-homing peptides” or NRG (asparagine–glycine–arginine) peptides, which, upon binding to APN with high affinity, facilitate the effective inhibition of tumor-induced angiogenesis. The high specificity of APN/CD13 for newly formed vessels allows the tumor-selective administration of, e.g., cytotoxic compounds [10]. Similarly, APN/CD13 can in principle also be used for the molecular imaging of tumor masses and (neo)angiogenesis in animal models and also in patients [11,12,13,14,15,16,17,18]. Pharmacological inhibitors of APN/CD13 have been proven to reduce tumor growth and progression in a variety of APN/CD13-positive tumors [10]. The inhibition of angiogenesis (or “tube formation” in vitro) is apparently only one of the underlying mechanisms here, because the therapeutic effects of APN inhibitors (ubenimex and actinonin) which do not depend on neoangiogenesis have also been described for acute or chronic myeloid leukemia (AML and CML) [19].

A major limitation of many currently used chemotherapeutic protocols, affecting long-term remission, is that resistant cancer stem cells (CSC) cannot be reached by those therapeutics. More recent publications have identified CD13 as a CSC-specific membrane marker, including for hepatocellular carcinoma (HCC), and cholangiocarcinoma [20,21]; it exhibits preferential expression on cells of the side population (SP), which are characterized by extremely high chemoresistance and tumorigenicity in transplantation models. From a functional point of view, CD13 is known to protect CSC from apoptosis [22]. Results from recent publications indicate that by pharmacologically inhibiting APN/CD13 enzymatic activity, the antitumor effects of chemotherapeutic agents, e.g., 5-fluorouracil, can be enhanced [10,23,24].

APN/CD13 has notably been linked to cell migration; the underlying mechanisms have largely been determined and are described here. Accordingly, pharmacological or antibody-mediated inhibition of the enzymatic activity of APN/CD13 has been shown to inhibit migration and invasion in various cancer models. CD13-regulated signaling pathways include extracellular signal-regulated kinase 1/2 (Erk1/2), phosphatidylinositol-4,5-bisphosphate 3-kinase (PI3K), and Wnt. Their contribution to migration/metastasis is reviewed here in depth.

## 2. Physiology and Pathology of APN/CD13

Aminopeptidase N (APN/CD13, EC 3.4.11.2) is a 150 kDa type II membrane Zn^2+^-dependent metalloprotease that has been identified as the leukocyte surface differentiation antigen CD13 [1]. The membrane-bound form exists as a homodimer and is directed to the apical cell membrane of polarized cells by an apical sorting signal in the catalytic ectodomain of APN/CD13 [25,26]. A slightly shorter but still functional soluble form can be generated via matrix metallopeptidase (MMP) 14-mediated shedding [27], and has been described as occurring in, e.g., serum or synovial fluid [28]. The ectopeptidase preferentially cleaves neutral amino acids from the N-terminus of oligopeptides [3]. APN, through its enzymatic activity, is involved in the degradation/modulation of several biologically active (neuro-) peptides, including met- and leu-enkephalin, oxytocin, vasopressin, angiotensin III, and immunomodulatory peptides such as tuftsin. Thereby, APN/CD13 has been implicated in, e.g., blood pressure regulation and nociception, but also in cell proliferation, migration, invasion, and metastasis as well as in angiogenesis. In addition to its peptidase activity, many cellular effects can be attributed to the non-enzymatic functions of APN/CD13; this is why APN is called a moonlighting enzyme (Figure 1) [3]. The non-enzymatic functions of APN/CD13 include the activation of signal transduction pathways upon ligandation of APN/CD13 [29,30,31,32,33], its role as an entry receptor for human coronavirus 229E (HCov-229E) [34], human transmissible gastroenteritis virus (TGEV) [35], and cytomegalovirus (HCMV) [36], as well as, e.g., cell–cell or cell–matrix adhesion [37], cell trafficking [29], endothelial invasion [37], and the maintenance of satellite stem cell populations [38].

APN/CD13 is strongly expressed in cells of the myelomonocytic lineage and is used accordingly as a routine marker in the diagnosis of lymphoma/leukemia. Mature and resting T cells lack detectable expression of APN/CD13. However, APN/CD13 expression in T cells has been observed in cancer and in response to T-cell activation [39,40,41,42,43,44,45,46], and has been associated with the suppressive activity of regulatory T cells [47]. The available data demonstrate that the inhibition of APN/CD13 enzymatic activity exerts strong immunomodulatory effects, mediated by—among other factors—induction of the expression of immunosuppressive cytokines such as transforming growth factor-β1 (TGF-β1) and interleukin-10 (IL-10), suppression of the production and release of stimulatory or pro-inflammatory cytokines, including IL-2 and interferon γ (IFNγ), and/or the support of regulatory T-cell function [48,49,50]. APN/CD13, as a moonlighting enzyme [3], has also been shown to exert a number of functions independently of its enzymatic activity, e.g., via Src-mediated phosphorylation of its cytoplasmic tail upon CD13 cross-linking [29], regulating components of membrane-tethered signaling complexes or monocytic/endothelial adhesion (Figure 1) [51]. This APN-mediated complex formation has been demonstrated to regulate, e.g., cell–cell and cell–extracellular matrix (ECM) adhesion [38,52,53,54], migration [53,55], endocytosis [31], antigen uptake [30], and myeloid cell–cell fusion [56].

## 3. Mechanistic Understanding of APN/CD13 in Cancer

### 3.1. Correlations and Prognosis in Cancers

Altered, and predominantly increased, expression of APN/CD13 has been shown for many tumor entities, including both hematological and solid tumors, as well as neighboring stromal cells [57]. There are data suggesting the existence of a cancer-associated APN/CD13 which may be structurally distinguished from APN/CD13 present in “normal” cells [58,59]. “Cancer” APN/CD13 may also show a change in substrate specificity compared with the “normal” enzyme. The alterations observed with the use APN/CD13-binding antibodies occur due to changes in glycosylation and may cause variable masking of protein epitopes [59].

Multiple studies have investigated the prognosis of patients with APN/CD13-positive tumors in various cancers. Tokuhara et al. showed that patients diagnosed with non-small cell lung cancer whose tumors are APN/CD13-positive have a lower 5-year survival rate than those with APN/CD13-negative tumors [8]. The average 5-year survival rate in patients with APN/CD13 expression has also been reported to be considerably lower than that in patients without the expression of APN/CD13, in a variety of cancers such as colon cancer [60] and osteosarcomas [5]. Increased tumor cell expression or elevated serum activity of APN/CD13 has been associated with poor prognosis in, e.g., pancreatic, hepatocellular [9], non-small cell lung cancer [6], breast cancer [61], Philadelphia chromosome-negative B-cell acute lymphoblastic leukemia [62], and multiple myeloma [7]. However, for squamous cell/adenosquamous carcinoma (SC/ASC) forms of gallbladder cancer, meningioma, and gastric cancer, a negative correlation between overall survival and the expression of APN/CD13 has been described [4,63,64].

In xenograft carcinoma animal models, the inhibition of APN/CD13 enzymatic activity reduces tumor growth.

There is a significant correlation between APN/CD13 activity and the volume of solid tumors, including colorectal, lung, and osteosarcoma [5]. Van Hensbergen et al. also reported a clear association between tumor load and soluble APN/CD13 (sCD13) activity in plasma [65].

### 3.2. APN/CD13 and Cancer Stem Cells (CSC)

A serious limitation of chemotherapeutic protocols that are currently applied in cancer therapy, which involve longer-term remission, is that so-called tumor stem cells (cancer stem cells—CSCs) escape the effects of therapeutic agents. CSCs exhibit a progenitor cell-like phenotype, are typical constituents of most tumors and cell lines derived thereof, and are able to self-renew. This clinically relevant ability supports the initiation and progression of the tumor, as well as its resistance, recurrence, and metastasis after therapy [66]. Due to their capacity for multilineage differentiation, CSCs may give rise to all cell types that are present in a given tumor sample. Resistance to chemotherapeutic CSC drugs is achieved through their increased expression of highly efficient efflux pumps [67]. CSCs can be both quantified and enriched via “sorting” [68,69]. An established FACS technique uses the rapid ejection of, e.g., Hoechst 33342, from progenitor cells by means of efficient efflux pumps, where CSCs present as a so-called side population (SP).

The CSC phenotype is determined by, among other features, the expression of stem cell surface markers. As with CSCs and non-tumor cells, CSCs and mature tumor cells undergo continuous transition, a process that enables expansion of tumor mass and contributes to the indefinite maintenance of tumor cell lines [70].

To overcome CSC “stemness”, overcoming CSC “therapy resistance” is considered a promising approach. As a proof of concept, this has already been successfully demonstrated in the targeting of the hepatic stem cell marker CD13 [24,71,72,73]. Recent publications demonstrate that APN/CD13 inhibitors sensitize the CSC to chemotherapeutic agents in colorectal and hepatocellular carcinoma [24,72,73,74].

CD13 has been identified as a mesenchymal stem cell, as well as a CSC-specific surface marker for, e.g., hepatocellular carcinoma and cholangiocarcinoma [75,76]. Cells of the side population (SP), which are characterized by their high chemoresistance and tumorigenicity in transplantation models [77,78], exhibit strong and preferential expression of CD13. Mechanistically, CD13 is associated with a lower extent of ROS-induced DNA damage after genotoxic stress. Therefore, APN/CD13 protects CSCs from apoptosis [78]. In hepatic cancer cells, TGF-β-induced epithelial-to-mesenchymal transition (EMT) is associated with elevated expression of CD13. Again, APN/CD13 functions by preventing a further increase in ROS levels and thereby supports CSC survival [79]. The pharmacological inhibition of APN/CD13 forces CSCs into apoptosis [79] and makes cells more sensitive to the effects of chemotherapeutic drugs, e.g., 5-fluorouracil, doxorubicin, or TNFα [73,74,80], which is why APN/CD13 inhibitors are considered cancer chemosensitizers [81].

CD13 has been shown to induce multi-drug resistance (MDR) in tumor cells [72]. Multi-drug resistance (MDR) is a phenomenon that renders certain cells resistant to the action of chemotherapeutic drugs. MDR is regarded as a typical feature of stem cells. In most settings, MDR is caused by the induced expression of efflux transporters. In CSCs, for example, dysregulated HH signaling facilitates the induction of ABC transporter expression via CD13 [82]. In the hedgehog pathway, APN/CD13 acts as a pseudoligand of the receptor PATCHED, and thereby sensitizes the pathway. This leads to the enhanced expression of ABCB1, ABCC2 (MRP2), ABCC3 (MRP3), and ABCG2 (P-glycoprotein), which together trigger an increase in drug resistance [21].

The tumor niche also crucially contributes to hematopoietic (HSC) and mesenchymal stem cell (MSC) maintenance and the resistance of disseminated tumor cells and CSCs to chemo- or radiation therapy. It has been clearly shown that bone provides niches that maintain the quiescence of CSCs, HSCs, and MSCs [83]. It has also been convincingly demonstrated that tumor cells disseminate into the bone even before a primary tumor can be detected. Therefore, tumor niches contribute to CSCs’ long-term survival, dormancy, and therapy resistance. Aging is associated with increased incidence of metastatic relapse. Singh et al. demonstrated age-dependent changes in the bone secretome that turn low- or non-proliferating dormant stem cells into proliferating cells. The PDGF-mediated expansion of pericytes has been identified as an underlying mechanism in this process [83].

Similarly, in HCC it has been shown that dormant CD13^+^ CSC residing in hypoxic areas of the tumor survive radiation or chemotherapy [23,77]. Compared with CD13^−^ cells, which undergo ROS-mediated apoptotic cell death following chemotherapy, dormant CSCs showed significantly fewer ROS-induced double-stranded breaks (DSB) [84].

The coexistence of dormant and proliferating CSCs also contributes to chemoradiation resistance in gastrointestinal cancer [85], whereas CD13^−^/CD90^+^ liver CSCs were shown to be sensitive to chemotherapy, and dormant CD13^+^/CD90^−^ cells have low proliferation ability, can survive in hypoxic areas with low ROS levels, and exhibit high metastatic activity [85]. Non-homologous end-joining, an error-prone repair mechanism, is dominant in dormant cells. Dormancy, therefore, has been proposed to represent an essential mechanism in CSC therapy resistance.

### 3.3. APN/CD13 Is Indispensable for Cellular Migration/Invasion

APN/CD13 is involved in a variety of biological processes, including cell survival, tumor cell invasion, metastasis, angiogenesis, resistance to anticancer agents, and relapse of cancer [3]. Given the role of APN/CD13 in migration, discussed above, it is not surprising that inhibition of the enzymatic activity of APN/CD13 has been shown to inhibit cell migration and/or invasion in a wide variety of cellular systems. Development of APN inhibitors such as antibodies, peptides, and non-peptidic molecules could be a promising therapeutic approach to the treatment of cancer. APN inhibitors are classified into two main categories according to their structure: amino-acid-based inhibitors and non-amino-acid-based inhibitors [86].

The aminopeptidase inhibitor bestatin has frequently been used in migration/invasion assays. It was originally isolated from Streptomyces olivoreticuli [87], and has been reported as an APN inhibitor in various types of malignancy, including colon adenocarcinoma [88], fibrosarcoma [89], melanoma [89], and renal cell carcinoma [89].

Bestatin, [(2S,3R)-3-amino-2-hydroxy-4-phenylbutanoyl]-(S)-leucine (trade name: ubenimex), was reported to competitively inhibit aminopeptidase B (EC 3.4.11.6), aminopeptidase N (EC 3.4.11.2), leucine aminopeptidase (EC 3.4.11.1), X-Trp aminopeptidase (EC 3.4.11.16), and leukotriene A4 hydrolase (EC 3.3.2.6) [87,90,91,92]. Although bestatin is a relatively unspecific aminopeptidase inhibitor, it has been used for several years, mostly experimentally and in the treatment of acute myelocytic leukemia [19]. However, it remains to be convincingly proven that the inhibition of APN/CD13 has any effect on cellular migration, since it was demonstrated that similar inhibitory effects on migration/invasion could be provoked by either aminopeptidase inhibition or administration of anti-CD13-monoclonal antibodies, or the genetic manipulation of APN/CD13 (knockdown, knockout, or overexpression), respectively.

Along these lines, in a transwell migration assay the motility/invasion of intraosseous malignant meningioma-Lee (IOMM-Lee) meningioma cells was significantly inhibited both by APN/CD13-siRNA and by the aminopeptidase inhibitor actinonin [64]. Furthermore, the results of earlier work demonstrated that the invasion of metastatic fibrosarcoma, melanoma, and renal cell carcinoma in Matrigel could be dose-dependently inhibited by the anti-CD13 monoclonal antibody clone WM15 [89]. WM15 has been shown to bind in close vicinity to the catalytic HELAH motif of APN/CD13 and thus partially inhibit its enzymatic activity [93].

It has also been reported that enhancing the length of the peptide chain in bestatin analogs can accelerate the binding process and generate more efficient APN inhibitors. Tosedostat is a derivative of bestatin, and is reported to be 300 times more potent than bestatin. It has been evaluated in clinical phases 1 and 2 for the treatment of acute myeloid leukemia, pancreatic cancer, and non-small cell lung cancer [86].

Numerous non-amino acid APN inhibitors, including phosphonic, phosphinic, and boronic acids, mercaptan, pyrrolidine, 3-amino-2-tetralone, β-amino-α-hydroxyl-phenylbutanoic acid (AHPA), indoline-2,3-dione, and cyclic imide have been identified as efficient APN inhibitors [86].

The key role that APN/CD13 plays in the process of cellular migration has been shown in the pioneering work of the laboratory of Linda H. Shapiro [53]. In migrating cells, phosphorylated APN/CD13 at the leading edge builds a complex with IQ motif-containing GTPase-activating protein 1 (IQGAP1), GTP-activated ADP ribosylation factor 6 (ARF6), the adenosine diphosphate ribosylation factor 6-guanine exchange factor (ARF6-GEF), Pleckstrin, and Sec7 domain-containing protein (PSD, syn: EFA6). This facilitates the expression of integrin α1 expression on the cell surface, in contrast to the endosomal accumulation and degradation of α1 integrin in cells lacking APN/CD13. ARF6-specific exchange factor 6 (EFA6), such as ARF6-GEF, needs to be localized at the plasma membrane (Figure 1). It is noteworthy that the membrane localization of IQGAP1 and EFA6 is lost in the absence of APN/CD13. Among other molecules, integrin α1 contributes to the establishment and regulation of cell adhesion to the extracellular matrix or neighboring cells. Cell adhesion is regarded as a key step during cell migration [94].

Bestatin was shown to inhibit the Ala-pNA hydrolyzing activity of MG63 and U-2 OS osteosarcoma cells, and this was associated with the inhibition of both migration and invasion [95]. In gastric cancer, the elevated expression of APN/CD13 is associated with resistance to cisplatin (CDDP) [80]. The pharmacological inhibition of APN/CD13 by bestatin caused CDDP-resistant gastric cancer cells to become sensitive to CDDP and was able to inhibit epithelial-to-mesenchymal transition (EMT), migration, and invasion [80]. Mechanistically, it has been shown that bestatin downregulates the expression of epithelial membrane protein 3 (EMP3), thereby weakening the APN/CD13/EMP3/PI3K (PI3K9/AKT/nuclear factor kappa B (NF-κB) axis [80] that promotes CDDP resistance.

Bestatin and more specific inhibitors of APN/CD13 have frequently been reported to reduce resistance to radiation or chemotherapy in vitro and in vivo. Tsukamoto et al. [96] reported that bestatin functions as a radiosensitizer both in vitro and in vivo, and enhanced the effect of radiotherapy on cervical cancer. The authors reported that bestatin can increase radiation-induced apoptosis by inhibiting APN/CD13 activity [96], and may therefore be proposed as a promising therapeutic approach to increase the efficacy of radiotherapy in cancers. Another study indicated that APN/CD13 can reduce sensitivity to paclitaxel in ovarian carcinoma. A negative association was reported between the expression of APN/CD13 and paclitaxel chemosensitivity in different cancer cell lines. The inhibition of APN/CD13 expression by means of APN/CD13 inhibition through the addition of bestatin or by using the siRNA technique can substantially increase chemosensitivity to paclitaxel in APN/CD13-expressing ovarian carcinoma cells. Thus, in combination with chemotherapy, APN/CD13 can be an appropriate target for carcinoma treatment [97]. It has been shown using HCC as an example that this strategy can, in principle, be implemented in patients [24,73,80].

Recent work has confirmed the ability of bestatin to suppress the migration and invasion of four different gastric cancer lines [98]. The authors identified NGFI-A binding protein 1 (NAB1), Erk1, and mitogen-activated protein kinase 3 (MAPK3) as mediators involved downstream of APN/CD13. Notably, the inhibition of APN/CD13 was reported to enhance Erk1/2 mRNA levels and the Erk1/2 activation (phosphorylation) of KARPAS-299 cells (Figure 1) [99]. Anti-CD13 cross-linking antibodies induce the clustering of CD13, tyrosine phosphorylation of CD13, and activation/phosphorylation of Src kinase, which in turn activates/phosphorylates focal adhesion kinase (FAK), mitogen-activated protein kinase 1 (MAPK1, ERK), and other components of the MAPK and phosphatidylinositol 3-kinase (PI3K) pathways. Furthermore, APN/CD13 regulates internalization of TLR4 and downstream innate signaling cascades via NF-κB and endosomal signaling via interferon regulatory factor 3 (IRF3) [31] (Figure 1).

APN/CD13 has been shown to control endothelial cell invasion in response to bradykinin. The binding of bradykinin to its receptor, B2R, facilitated signal transduction and, subsequently, receptor internalization (Figure 1) [100,101]. The inhibition of APN enzymatic activity prevented B2R internalization, modified downstream signaling such as cdc42 activation and filopodia formation, and thereby inhibited cell migration [37,100,101]. APN/CD13 influences the recruitment and migration of lymphocytes, e.g., to adopt an inflammatory focus, by proteolytically processing the chemotactic antiangiogenetic active chemokine, and CXC motif chemokine ligand 11 (CXCL11). Especially in combination with dipeptidyl peptidase 4 (DPP4)/CD26, N-terminally truncated forms of CXCL11 are formed by APN/CD13; they bind and desensitize CXC motif chemokine receptor (CXCR3) and CXCR7 chemokine receptors, thereby reducing lymphocyte infiltration and increasing endothelial cell migration [102].

APN/CD13 also promotes HCC progression and contributes to sorafenib resistance. Similar to the pharmacological inhibition of APN/CD13, described above for gastric cancer, both the administration of bestatin and the siRNA-mediated knockdown of APN/CD13 overcome HCC resistance to sorafenib. The knockdown of APN/CD13 has been demonstrated to decrease the Matrigel invasion of MHCC97H and HCCLM3 cells, whereas the overexpression of APN/CD13 enhanced the invasion of Huh7 cells [32]. The modulation of APN/CD13 consistently affected the activity of the NF-κB-pathway. Mechanistically, APN/CD13 has been shown to increase the protein stability of histone deacetylase 5 (HDAC5), which becomes increasingly ubiquitinated to undergo proteasomal degradation upon the knockdown of APN/CD13 in various HCC cell lines [32]. The increased stability of HDAC5 in turn has been shown to stabilize lysine demethylase 1 (LSD1), leading to the demethylation of p65, and thus the activation of NF-κB signaling. The study identified the CD13/HDAC5/LSD1/NF-κB axis as a promising new therapeutic target for the treatment of HCC (Figure 1) [32].

Bestatin also shows inhibitory activity in the migration and invasion of the renal carcinoma lines 786-O and OS-RC-2 [103], the bladder cancer cell lines RT112 and 5637, and the osteosarcoma cell lines Mg63 and U 2-OS.

The anti-CD13 monoclonal antibody clone WM15 not only reduced the APN/CD13 enzymatic activity of neutrophils and recombinant human APN/CD13, but also inhibited IL-8-induced neutrophil migration through collagen I [104].

The inhibition of APN/CD13 impairs migration in the great majority of cell lines. It has been clearly established experimentally that APN/CD13 participates in inflammatory cell trafficking. Circulating monocytes continuously patrol for inflammatory adhesion molecules to enter tissues, where they produce cytokines to recruit monocytes/inflammatory cells to sites of injury/infection. In the murine model of thioglycollate-induced peritonites, monocyte trafficking has been shown to depend on both endothelial and monocytic APN/CD13 expression: APN/CD13 knockout mice exhibited significantly lower peritoneal cell counts of monocytes, macrophages, and dendritic cells [52]. The authors further demonstrated that it the APN/CD13 C-terminal part determines monocyte/endothelial adhesion and cell trafficking.

Likewise, myeloid-derived suppressor cells (MO-MDSC) transmigrate less effectively than monocytes through hepatic endothelial monolayers [105]. MO-MDCSs notably exhibit lower expression of APN/CD13 than monocytes, and their transmigration through the liver endothelium is reduced by the inhibition of APN/CD13 activity [105].

Bestatin inhibits the migration and invasion of bladder cancer cell line 5637, of renal cell carcinoma cell lines 786-O and OS-RC2, of gastric cancer cell lines MKN-28, MGC-803, BGC-823, and SGC-790, and of osteosarcoma cell lines MG-23 and U2-OS [95,98,103,106]. In all published studies on this topic, ubenimex was also able to reduce invasion. It was recently shown that bestatin furthermore inhibits the transmigration of human monocytic myeloid-derived suppressor cells, which are able to suppress inflammation but may cause impaired immune surveillance in cancer [107].

Similarly to bestatin, the inhibitory anti-CD13 monoclonal antibody WM15 was also able to inhibit transmigration, whereas the non-inhibiting or more importantly non-cross-linking clones SJ1D1 and 3D8 showed no such effect.

Similar inhibitory effects on migration can be also achieved by more specific inhibitors of APN/CD13. For example, the bestatin-derived inhibitor BE15, which shows higher inhibitory activity towards APN/CD13 compared with bestatin, strongly inhibited migration, capillary tube formation, and invasion [108].

### 3.4. Role of APN/CD13 in Angiogenesis

APN/CD13 has been implicated in angiogenesis both in health and disease, e.g., cancer. Angiogenic stimuli such as hypoxia and increased growth factor concentrations can also cause the strong induction of APN/CD13 expression on endothelial cells [10]. This up-regulated APN/CD13 was shown to be functionally relevant and a requirement for new vessel formation: the application of an inhibitor of APN/CD13, bestatin, or inhibitory monoclonal anti-CD13 antibody My7 severely impaired the tube formation of human umbilical vein endothelial cells (HUVECs) on Matrigel, without compromising cell proliferation [10]. Stimuli known to promote neoangiogenesis/vessel formation in, e.g., cancer, include hypoxia and ischemia; among other effects, these induce the expression of angiogenic growth factors [109], which in turn activate the quiescent endothelial cells of existing vessels to proliferate and migrate toward the tumor tissue [110].

It has been demonstrated that peptides containing the NGR motif bind exclusively to endothelial cells of angiogenic but not normal vasculature [10]. APN/CD13 has been identified as an NGR receptor; meanwhile, the use of radiolabeled NGR peptides, as well as cytotoxic or antiangiogenic substances such as doxorubicin, 5′-fluorouracil, coagulase, or tissue factor coupled with NGR motifs represents a widely applied principle in CD13-targeted experimental tumor therapy [15,16,17,111,112] (which has already been applied to humans [11]), of tumor imaging [14]. NGR peptide-directed selective vascular targeting through the specific delivery of tumor necrosis factor-alpha (TNFα) has also been the subject of phase I and phase II studies [12,13,18] (Figure 1).

A HUVEC capillary tube formation assay was applied when the key role of APN/CD13 in angiogenesis was first described [10,113], and the method has since been widely used as a reliable in vitro model of angiogenesis. It was thereby shown that pharmacological inhibitors of APN/CD13, which include highly potent, selective inhibitors, severely impair tube formation [114,115,116]. In order to eliminate doubt about the specificity of the inhibitors, the effects of the APN/CD13 inhibitors were partially complemented with and directly compared to those obtained with anti-CD13 monoclonal antibodies [60,117] or APN/CD13 siRNA [118]. The determination of intratumor microvessel density and the dorsal yolk sac assay have been employed as appropriate alternatives to the HUVEC tube formation assay [60,114]. Conflicting data exist concerning the question of whether aminopeptidase enzymatic activity is required for angiogenesis. The results of some studies indicate that APN enzymatic activity might be involved in angiogenesis. In support of this view, inhibitory antibodies such as clone WM15, but not non-inhibiting antibodies, compromised tube formation [114]. In a landmark study by Linda H. Shapiro’s group [113], the partially inhibiting monoclonal antibody clone My7 inhibited capillary tube formation, which might point to the involvement of enzymatic activity. In accordance with the role of APN/CD13 as a key regulator of angiogenesis, hypoxia and angiogenic factors such as vascular endothelial growth factor (VEGF) have been shown to increase the mRNA and surface expression of APN/CD13 on primary human endothelial cells [113].

### 3.5. APN/CD13 in Metastasis

As mentioned above, APN/CD13 is frequently expressed at elevated levels in tumors or the tumor vasculature. This fact has been exploited for targeted drug delivery with so-called tumor-homing peptides (NRG peptides). These peptides show specific high-affinity binding to APN/CD13. Thus, APN/CD13 represents a promising target for the inhibition of tumor-induced angiogenesis, as well as for the tumor-selective administration of cytotoxic substances [10,15,112,119,120]. From this perspective, APN/CD13 has been and still is used for determining the mass, activity, and extent of (neo)angiogenesis and the presence and location of tumor metastasis in animal models, as well as in human patients [121,122,123,124].

Tumor invasion is a complex process consisting of cell adhesion and migration, and the (proteolytic) degradation of tissue barriers and extracellular matrix/matricellular proteins. An anti-CD13 monoclonal antibody, WM15, has also been shown to inhibit the degradation of type IV collagen by SN12M cells in a concentration-dependent manner, as was observed previously by applying the aminopeptidase inhibitor bestatin [125]. Similarly, the overexpression of basic fibroblast growth factor (bFGF) isoforms in IF6 melanoma cells, which led to a strong increase in APN/CD13 expression/activity, enhanced cells’ invasion through Matrigel [126].

APN/CD13 expression is elevated in HCC, and HCC cell lines differ in the extent of their APN/CD13 expression. High-APN/CD13 HCC cells have greater metastatic activity than low-APN/CD13 HCC cells [127]. The growth of HCC cells was substantially reduced in vitro and in vivo upon the knockout of APN/CD13. This was accompanied by the reduced migration and invasion of APN/CD13 knockout cells, whereas cells with overexpressed APN/CD13 exhibit elevated migration/invasion [127]. Accordingly, in liver tumor metastasis models, APN/CD13 knockout mice showed fewer and smaller intrahepatic metastatic nodules, lower incidence of tumors spreading to other organs outside the liver, and longer survival. Mechanistically, APN/CD13 knockout caused the dephosphorylation of Erk1/2. It can be concluded, therefore, that the MAPK/ERK signaling pathway is crucial for APN/CD13-mediated HCC growth and metastasis, and that Erk acts downstream of APN/CD13. Phosphoproteome analysis found that branched-chain α-keto acid dehydrogenase kinase (BCKDK) serine 31 is most strongly regulated (dephosphorylated) under conditions of APN/CD13 knockout, and that BCKDK can increase ERK phosphorylation in HCC cells. This effect is dependent on the presence of APN/CD13 [127] (Figure 1).

In carcinoma gallbladder adenocarcinoma, there was a marked reduction in APN/CD13 compared with normal tissue [63]. The lack of APN/CD13 was significantly correlated with larger tumors, higher tumor–node–metastasis (TNM) staging, invasion, metastasis to regional lymph nodes, and ineligibility for surgical resection. APN/CD13 negativity was also correlated with decreased overall survival in patients with the disease [63].

It has recently been identified that glycosylated CD13 expressed in breast cancer binds E-selectin, and it has been suggested that it plays a role in the development of metastasis [128].

### 3.6. Soluble APN/CD13

Soluble APN/CD13 has been shown to retain its enzymatic activity [129] and is detectable in blood and other biological fluids. In the synovial fluid of patients with rheumatoid arthritis, soluble ANP has been shown to exert arthritogenic properties and to retain the chemoattractant activity of APN/CD13 towards mononuclear and T cells [27,130]. Mechanistically, it was recently shown that soluble APN/CD13 exerts arthritogenic, chemoattractant, and angiogenic effects upon direct binding to, and subsequent signaling through, the bradykinin 1 receptor (B1R) [28]. The effects are probably independent of APN/CD13 enzymatic activity [130] (Figure 1).

However, BR antagonists were shown to inhibit directly APN/CD13 enzymatic activity [100,101]. It was previously shown that chemotactic activity is mediated by a G-protein-coupled receptor [130]. Contributing to the pathology of rheumatoid arthritis (RA), soluble APN/CD13 induced the chemotaxis of mononuclear cells/macrophages [131] and cytokine-activated T cells, with a T-cell population similar to that found in the RA synovium [130]. Interestingly, the expression of APN/CD13 in T cells has been described as induced upon contact with fibroblast-like synoviocytes (FLS) and is associated with T-cell activation [44,132]. The immunodepletion of APN/CD13 and the inhibition of APN/CD13 enzymatic activity both reduce the growth and migration of FLS [121].

## 4. Clinical Insights and Application Challenges

As already mentioned, APN/CD13 plays a key role in many physiological and pathological processes including angiogenesis, immune response, tumor invasion, and metastasis. Thus, APN has become an attractive target for pharmacological intervention in cancer. As highlighted in recent reviews [57,58,133], the potential of APN/CD13 as a target for tumor therapy has prompted a series of pre-clinical and clinical studies systematically investigating the effects of the inhibitors/ligands of APN/CD13, alone or in combination. However, due to the overlapping substrate specificity of the aminopeptidase family, the development of specific inhibitors that target APN/CD13 is yet to be fully achieved and remains an important challenge [58].

Furthermore, much effort remains necessary to identify the tumor entities that are most susceptible to APN/CD13-based drug targeting or combinatory therapy employing APN/CD13 inhibitors/ligands. This is particularly true for protocols that aim to overcome CSC resistance to therapy and tumor relapse.

Metal-chelating inhibitors such as bestatin, amastatin, and actinonin were among the first APN/CD13 inhibitors to be developed. Bestatin, an inhibitor of aminopeptidase, was first described in 1976 as being produced by actinomycetes [87]. Bestatin has been shown to exhibit potent anti-tumor, anti-angiogenic, and immunomodulatory effects in various preclinical studies of solid tumors, including malignant melanoma, and malignancies of the lung, stomach, and bladder [2]. Several clinical trials have been conducted to evaluate the efficacy of bestatin in the treatment of various cancers, including acute myeloid leukemia (AML) and non-small cell lung cancer (NSCLC). The results of a meta-analysis of these trials showed that a combination of bestatin with standard therapy can improve the survival rates of patients with these malignancies [134]. Wakita et al. randomized cases of patients diagnosed with AML who gained complete remission upon consolidation therapy with or without bestatin. They reported that overall survival and relapse-free survival increased in the group treated with bestatin [135]. In another placebo-controlled trial for squamous cell lung cancer, whereby the administration of bestatin and the placebo was randomized, significant enhancements in overall survival and relapse-free survival were indicated [136].

In Japan, bestatin has for more than two decades been employed as a chemotherapeutic agent, as maintenance treatment for patients suffering from myeloid leukemias or solid tumors [2,135,137]. However, it has not been approved by the European Medicines Agency (EMA) or the US Food and Drug Administration (FDA) [138].

Actinonin, a natural product-based inhibitor of APN/CD13, was first isolated from actinomycetes in 1985 [139]. It has been shown in various preclinical studies to exhibit potent anti-tumor, anti-angiogenic, and anti-inflammatory effects in vitro and in vivo [14,140]. Other inhibitors such as phebestin, which was discovered in 1997, are yet to be investigated in cancer [141].

Two decades after the discovery of bestatin, Krige et al. described tosedostat or CHR-2797 [142]. They demonstrated that bestatin and tosedostat can decrease the quantities of intracellular amino acids in tumor cells by blocking protein recycling, which can suppress tumor growth. In phase I/II trials, tosedostat, a cyclopentyl ester (CHR-2797), was found to be safe and effective in relapsed and refractory acute myeloid leukemia (AML), but adding it to standard chemotherapy negatively impacted the therapeutic outcomes of AML patients. A phase I study of CHR-2797 monotherapy showed tolerability and preliminary efficacy in a subset of patients with advanced solid tumors, indicating that further clinical investigation is needed [57]. Two clinical trials have been completed in which tosedostat was used against pancreatic ductal adenocarcinoma and myelodysplastic syndrome (NCT02352831 and NCT02452346, respectively) [57,58]. Tosedostat is an aminopeptidase inhibitor of the M1 metalloenzymes, a family of proteins containing peptidases with a zinc ion, which include APA, APN, APB, puromycin-sensitive aminopeptidase (PuSA), and leukotriene A4 hydrolase [143]; it has been described as a strong anti-proliferative, antiangiogenic, and proapoptotic agent [138,144]. Tosedostat is more potent than bestatin; it can inhibit cell proliferation 300 times more effectively [142]. Thus far, various clinical trials using tosedostat have been conducted.

The first research of tosedostat in humans (CHR-2797-001) involved 40 patients diagnosed with advanced solid tumors [145]. A phase Ib study (CHR-2797-003) that employed daily oral tosedostat in combination with paclitaxel was conducted in patients suffering from advanced or metastatic cancer with solid tumors; this was generally well-tolerated except for the high incidence of paclitaxel-related infusion reactions [146]. Another phase I/II trial including patients with stage IIIb, stage IV, or recurrent metastatic non-small cell lung cancer (NSCLC) was initiated to evaluate the combination of tosedostat and erlotinib in solid tumors (CHR-2797-005). However, this study was prematurely terminated due to limited participant enrolment, and no data analysis was conducted [143]. Subsequent studies have mostly concentrated on people suffering from acute myeloid leukemia (AML), myelodysplastic syndromes (MDS), and multiple myeloma. After the promising in vitro anti-leukemic activity of tosedostat, an initial phase I/II clinical trial (CHR-2797-002) in diagnosed elderly and/or previously treated adult patients was conducted. In the study, 16 patients received the drug in the phase I dose-escalation portions, and 41 patients participated in the phase II dose-expansion cohort. In conclusion, tosedostat was well tolerated, with 28% of patients with acute myeloid leukemia responding significantly, 14% achieving complete remission, and 14% attaining a partial response [147].

The OPAL study (CHR-2797-038) was a multicenter phase II study of tosedostat in 73 elderly patients with relapsed or resistant acute myeloid leukemia [148]. Eight patients who had previously been enrolled in the OPAL study were given extended treatment with tosedostat within the TOPAZ study (CHR-2797-045). To explore further the efficacy of tosedostat, the Dutch–Belgian Hemato-Oncology Cooperative Group (HOVON) initiated a randomized phase II multicenter study (HOVON 103). Following the initial OPAL results, which suggested that myelodysplastic syndrome patients who have undergone previous hypomethylating-agent therapy can benefit from tosedostat, a phase I/II single-institution study was launched to investigate the effects of tosedostat in combination with azacitidine or cytarabine in elderly patients with high-risk myelodysplastic syndrome or acute myeloid leukemia (NCT01636609) [143].

A randomized phase II study (NCT01567059) of daily tosedostat in combination with hypomethylating agents, such as cytarabine or decitabine, showed a complete remission or complete remission with incomplete recovery rate of 53% in newly diagnosed patients with acute myeloid leukemia or high-risk MDS [149].

In another study on older patients diagnosed with AML, tosedostat was added to standard chemotherapy and was reported to have detrimental effects on the treatment outcome [150].

APN/CD13 has been suggested as a therapeutic strategy for directing treatment against malignant cells. Different strategies have been applied to achieve this goal, including the use of (I) antibody–drug conjugates, (II) chimeric antigen receptor T cells, and (III) peptide–drug conjugates (NGR–peptide conjugates).

Antibody–drug conjugates (ADCs) consist of monoclonal antibodies (mAbs) specific to tumor cell surface antigens (e.g., APN/CD13) in conjugation with a powerful active anti-tumor agent [151]. ADCs interact with targeted surface antigens on tumor cells and become internalized in the cells, where toxic agents are released and lead to cell death [152]. APN/CD13 has been used as a target for an ADC therapeutic approach to directing MI130110, an anti-tumor therapeutic agent effective against tumor tissue. MI130110 is the conjugate of PM050489, a cytotoxic agent that binds tubulin and impairs microtubule dynamics, with an anti-CD13 monoclonal antibody. In vitro experiments revealed that the conjugate induced mitotic catastrophe, resulting in cell death. Furthermore, in vivo studies indicated considerable suppression of tumor growth in CD13^+^- fibrosarcoma mice [151].

Adoptive T-cell cancer therapy using chimeric antigen receptor (CAR)-expressing T cells has been successfully applied in the elimination of relapsed/refractory B-cell lymphoma and B-cell lymphocytic leukemia, as well as multiple myeloma, via the targeting of CD13 [153]. The CAR constructs feature an extracellular ectodomain composed of a single-chain variable fragment originating from a mAb, anchored to the cells via a transmembrane domain, followed by the intracellular costimulatory domains 4-1BB and/or CD28, and CD3ζ for signal transduction [154]. CAR T-cell therapy has not been approved for solid tumors. While CAR T cells particularly target tumor proteins, remarkably, their efficacy is related to the presence of appropriate mAbs for the CAR constructs. There are also various obstacles to generating effective CAR T cells capable of exerting cytotoxicity, such as accurate involvement of the T cell with the target tumor cell to bring about cell death. He et al. developed a sequentially tumor-selected antibody and antigen retrieval (STAR) system, which can quickly separate various nanobodies that have affinity for AML cells and enable CAR T cells with anti-AML effects. The STAR method was employed to isolate nanobody 157 attached to APN. APN CAR T cells effectively eliminated AML in vitro and in vivo. CAR T cells that target APN and TIM3, which are up-regulated in AML leukemia stem cells, have been found to reduce patient-derived AML in in vivo models. Decreased toxicity to human bone marrow stem cells and peripheral myeloid cells in vivo has also been reported, leading to the proposal of a promising method for CAR T-cell therapy in AML patients [153].

Melphalan, flufenamide, and melflufen are first-in-class anticancer peptide–drug conjugates (PDCs). The chemotherapeutic drug melphalan was discovered more than 50 years ago and has been widely used since [144]. Melflulfen is an FDA-approved medication for the treatment of multiple myeloma. This compound, considered a pro-drug activated by aminopeptidases expressed in multiple myelomas (such as APN, leucyl aminopeptidase 3 (LAP3), arginyl aminopeptidase (RNPEP), and leukotriene A4 hydrolase (LTA4H) [58,155]) has been studied in phase I/II (HORIZON, ANCHOR) [156,157] and phase III (OCEAN) [157,158] trials [57,58]. Melflufen is particularly active in plasma cells in relapsed/refractory multiple myeloma patients. In vitro and in vivo studies based on cytotoxicity assays have demonstrated that melflufen is preferable to melphalan or other alkylators in different types of tumors. Melflufen was proven to be more effective and to exhibit no more toxicity than melphalan [155]. Melflufen plus dexamethasone showed positive clinical efficacy and a tolerable safety profile in extensively pretreated patients with relapsed/refractory multiple myeloma, according to phase II single-arm trials (such as HORIZON). Moreover, HORIZON demonstrated that melflufen is effective in treating patients who are resistant to alkylators. The results of the phase III OCEAN study, which assessed the efficacy and safety of melflufen and dexamethasone against pomalidomide and dexamethasone in patients with relapsed/refractory multiple myeloma, revealed better progression-free survival with melflufen [158]. Additionally, ANCHOR (OP-104) showed that a triplet regimen of melflufen plus dexamethasone with daratumumab or bortezomib led to encouraging activity in patients with relapsed/refractory multiple myeloma who had received a high burden of prior treatment and had poor prognostic indicators.

Meanwhile, as mentioned above, the use of radiolabeled NGR peptides as well as cytotoxic or antiangiogenic substances such as doxorubicin, 5′-fluorouracil, coagulase, or tissue factors coupled to NGR motifs has become widespread in experimental tumor therapy [15,16,17,111,112] and tumor imaging [14]. This NGR peptide-dependent tumor-targeting method has also been the subject of clinical studies. A fusion protein in which an NGR peptide replaced the transmembrane domain of truncated tissue factor (tTF) resulted in tumor vascular thrombosis and occlusion with infarction. These fusion proteins were evaluated to determine their critical therapeutic characteristics in vitro and in vivo, and also in a phase I trial [11,156,157]. A total of 17 patients with advanced tumors participated in this first phase I clinical study, showing little risk of side effects and good tolerance to the drug. Tumor necrosis factor-α coupled with NGR (NGR-hTNF) is another NGR-peptide-targeted drug, and is in phase II/III clinical trials [13,159,160]. NGR-hTNF targets CD13^+^ vasculature and tumor necrosis factor provokes endothelial permeabilization. The use of R-CHOP (rituximab, cyclophosphamide, doxorubicin, vincristine, and prednisone) for therapy in patients with primary central nervous system lymphoma (PCNSL) is limited by the poor penetration of related drugs across the blood–brain barrier (BBB). Tumor necrosis factor-α coupled with NGR in combination with R-CHOP (NGR-hTNF/R-CHOP) improved the tumor access of cytotoxic drugs in patients with relapsed/refractory PCNSL. In phase II/III trials, the combination therapy was found to be effective and safe [12,13,160].

## 5. Open Questions and Future Directions

As described above, the inhibition of APN/CD13 enzymatic activity sometimes results in biological effects similar to those of non-inhibitory ligands. This raises the question whether endogenous substrates of APN/CD13 induce similar or identical signaling pathways to some of the non-inhibitory ligands. However, endogenous substrates possibly remain to be identified, in particular, endogenous substrates that contribute substantially to the regulation of migration and invasion. However, it cannot be ruled out that certain substances act primarily via the liganding of APN/CD13 despite their enzyme-inhibiting activity.

Data suggest that APN/CD13 in tumor cells differs in structure from APN/CD13 in “normal” cells and may also have unique substrate specificity. This needs to be followed up in future work, through the application of suitable cancer models. This may lead to essential improvements in therapeutic applications, by means of APN/CD13 targeting.

Inhibitors of APN/CD13 enzymatic activity have in numerous protocols been proven to increase the efficacy of other chemotherapeutic compounds. One possible mechanism is the loss of the stem cell properties of CSCs and better accessibility of the resulting differentiated cells. However, researchers are yet to establish fully which chemotherapeutic agents can be most effectively potentiated by APN/CD13 inhibitors and which tumor entities are the most suitable for sensitization via APN/CD13. However, the effects of APN/CD13 inhibitors, alone or in combination, on stem cell numbers, CSCs’ drug sensitivity, or CSC differentiation have not yet been studied in detail.

In various settings, primarily in the context of immunomodulation and the treatment of autoimmune diseases, the dual inhibition of APN/CD13 and of the ectopeptidase DPP4/CD26 has been shown to be more effective than the inhibition of APN/CD13 alone. In this context, dual-specific inhibitors of APN/DPP4 have been developed and have been proven to be effective in vitro and in vivo, e.g., in mouse models of multiple sclerosis or stroke [48,50,161]. In the latter, dual inhibition of APN/DPP4 supported neuronal survival after cerebral ischemia, as was concluded from the substantial reductions in cortical lesions after cerebral ischemia [161]. The molecular mechanisms underlying these neuroprotective effects remain to be fully elucidated.

The possible synergistic effects of the dual inhibition of APN/CD13 and DPP4/CD26 on migration and invasion are also yet to be studied.

## 6. Conclusions

APN/CD13 has enzymatic and non-enzymatic functions. The expression and activity of APN/CD13 are altered in many tumors, and this is typically of great relevance to patients’ prognosis and survival. APN/CD13 is indispensable for cellular migration, invasion, and metastasis. Various signal-transduction pathways, some of which are very well understood, contribute to the development of the migratory phenotype of a cell.

## Figures and Tables

**Figure 1 biomedicines-11-00724-f001:**
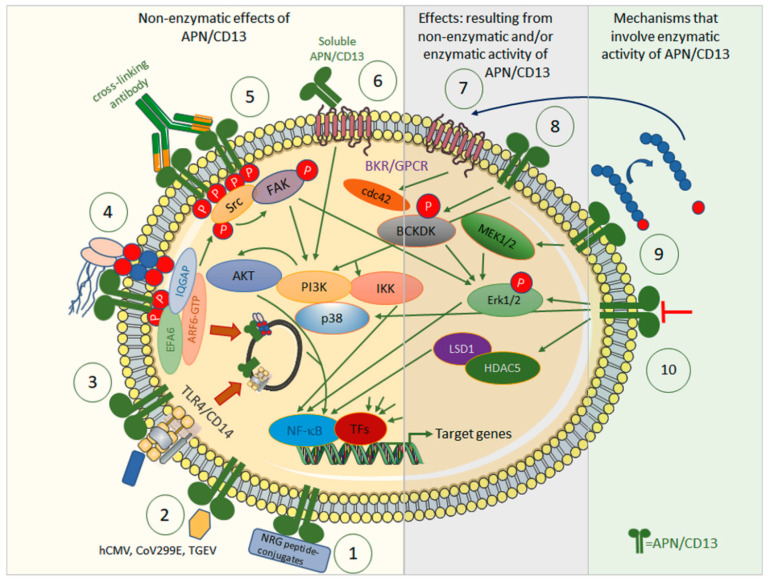
Effects of APN/CD13 by means of its enzymatic activity (**right**) or independently of it (**left**), and the signal transduction pathways involved. Non-enzymatic: ① APN/CD13 allows tumor-specific drug targeting, with, e.g., NRG-peptide conjugates, against APN/CD13-positive tumors or associated cells. ② APN/CD13 serves as a receptor for certain human and porcine viruses, such as CoV229E, hCMV, and TGEV. ③ APN/CD13 regulates internalization of TLR4 and downstream innate signaling cascades via NF-κB and endosomal signaling via IRF-3. ④ In wild-type cells, APN/CD13 is internalized together with integrins and IQGAP-1 into early endosomes, sorted into recycling endosomes, and returned to the cell surface. Lack of APN/CD13 leads to transfer to lysosomes and degradation of integrin instead. Cross-linking of APN/CD13 leads to its clustering and tyrosine phosphorylation with subsequent activation of FAK and ERK kinases. APN/CD13 also tethers the IQGAP1-ARF6-EFA6 complex to the plasma membrane to promote ARF6 activation, β1 integrin recycling, and cell migration. ⑤ Cross-linking antibodies induce the clustering of CD13, tyrosine phosphorylation of CD13, and activation/phosphorylation of Src kinase, which in turn activates/phosphorylates both FAK and ERK kinases, and other components of the MAPK and PI3K pathways. ⑥ Strong chemoattractant, angiogenic, and arthritogenic activity has been attributed to soluble APN/CD13. Soluble APN/CD13 binds to bradykinin receptor 1 (B1R), which is highly expressed in, e.g., synovial fluid in rheumatoid arthritis. ⑦ APN/CD13 controls access of bradykinin to its bradykinin 2 receptor to induce small GTPase cdc42 signaling, filipodia formation, and thus, migration. ⑧ APN/CD3 mediate phosphorylation at serine 31 of branched chain alpha-ketoacid dehydrogenase kinase (BCKDK) to promote its interaction with and the activation/phosphorylation of ERK1/2. Enzymatic activity involved: ⑨ APN/CD13 regulates availability and/or receptor specificity of various bioactive peptides. These may contribute to altered cellular signaling via their specific receptors, e.g., B2R. ⑩ Inhibitors and (partially) inhibiting antibodies against APN/CD13 have been shown to influence, e.g., angiogenesis and migration. APN/CD13 binds to and stabilizes HDAC5. Inhibition or lack of APN/CD13 destabilizes HDAC5. APN/CD13 may also signal via p38 MAPK.

## Data Availability

Not applicable.

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
