# Peer review of "The Role of the Ectopeptidase APN/CD13 in Cancer"

_biomedicines, 2023, doi:10.3390/biomedicines11030724_

Round 1

Reviewer 1 Report

This is an interesting review of the potentially interesting molecule APN/CD13. The manuscript could be improved if the authors summarized more clearly what are the signaling pathways that are related to the enzymatic activity of APN and what are the enzymatic activity-independent signaling pathways. These should be included in two independent figures or two subfigures with a clear and concise legend. The existing Figure 1 with its legend has no connection to the text and does not efficiently summarize the existing knowledge.

The authors should also clearly describe what are the pharmacological interventions that are being studied about APN/CD13 in cancer and at what stage (preclinical/clinical etc) each potential therapeutic is. 

Finally, the authors should incorporate some more recent literature that is supplementary to their work, e.g. (but this is an example, not an exhaustive list)

Barnieh FM, Loadman PM, Falconer RA. Is tumour-expressed aminopeptidase N (APN/CD13) structurally and functionally unique? Biochim Biophys Acta Rev Cancer. 2021 Dec;1876(2):188641. doi: 10.1016/j.bbcan.2021.188641. 

Author Response

We would like to thank the reviewers and editors for a thorough review of our manuscript. With the help of the very helpful comments and advice, we believe that the manuscript has been significantly improved. Thank you very much!

Our response in detail to the points raised by the reviewers follows here:

Reviewer 1

This is an interesting review of the potentially interesting molecule APN/CD13. The manuscript could be improved if the authors summarized more clearly what are the signaling pathways that are related to the enzymatic activity of APN and what are the enzymatic activity-independent signaling pathways. These should be included in two independent figures or two subfigures with a clear and concise legend. The existing Figure 1 with its legend has no connection to the text and does not efficiently summarize the existing knowledge.

We are grateful for this comment. Although it is not always possible, to clearly separate non-enzymatic effects of APN/CD13 from enzymatic activity – dependent mechanisms we now present the most likely mode of action here. We thoroughly revised Figure 1 and include a new Figure1 in the revised version of the manuscript.

The authors should also clearly describe what are the pharmacological interventions that are being studied about APN/CD13 in cancer and at what stage (preclinical/clinical etc) each potential therapeutic is. 

We thank the reviewer for this comment. Recently, excellent reviews already addressed the point of clinical studies, this is why we omitted a chapter on this topic in the first version of the manuscript. As suggested, a new chapter and new references are added to the revised version of the manuscript.

Finally, the authors should incorporate some more recent literature that is supplementary to their work, e.g. (but this is an example, not an exhaustive list)

Barnieh FM, Loadman PM, Falconer RA. Is tumour-expressed aminopeptidase N (APN/CD13) structurally and functionally unique? Biochim Biophys Acta Rev Cancer. 2021 Dec;1876(2):188641. doi: 10.1016/j.bbcan.2021.188641. 

We thank the reviewer for pointing this weakness of the manuscript to us. The reference list has been significantly extended.

Reviewer 2 Report

APN/CD13 has already appeared as a potential prognosis marker for many years and a potential target for cancer therapy. In recent years, there have been many more crucial clinical and mechanistic insights into it. It is a very nice and concise review of the key findings about APN/CD13 and summarizes/proposes future perspectives. Also, the topic is timely.

In general, the review is informative, and the statements are mostly
accurate.
Comments:
1)    Please include more recent studies in the review.
2)     I would suggest following the structure for the main body: 1. physiology and pathology of APN/CD13 2. Mechanistic understanding of APN/CD13 in cancer 3. Clinical insights/ application challenges of APN/CD13
3)    Please expand on the mechanistic part  - the current description is somehow not very specific.
4)    Please include a few sentences on future directions.
5) APN/CD13 plays a role in angiogenesis, tumour angiogenesis and angiogenesis at metastatic sites, and vessels play an important role in maintaining dormancy or tumour cells and their reactivation PMID: 31292293; please add a section on this.

Author Response

Reviewer 2

APN/CD13 has already appeared as a potential prognosis marker for many years and a potential target for cancer therapy. In recent years, there have been many more crucial clinical and mechanistic insights into it. It is a very nice and concise review of the key findings about APN/CD13 and summarizes/proposes future perspectives. Also, the topic is timely.

In general, the review is informative, and the statements are mostly
accurate.
Comments:
1)    Please include more recent studies in the review.

We thank the reviewer for this comment. The reference list has been significantly extended, including more recent papers.

2)     I would suggest following the structure for the main body: 1. physiology and pathology of APN/CD13 2. Mechanistic understanding of APN/CD13 in cancer 3. Clinical insights/ application challenges of APN/CD13

The proposed structure has been applied to the revised version of the manuscript.

3)    Please expand on the mechanistic part  - the current description is somehow not very specific.

We are grateful for this comment and tried to be more clear now when it comes to mechanisms.

4)    Please include a few sentences on future directions.

We thank the reviewer for this comment. Changes have been made accordingly.

5) APN/CD13 plays a role in angiogenesis, tumour angiogenesis and angiogenesis at metastatic sites, and vessels play an important role in maintaining dormancy or tumour cells and their reactivation PMID: 31292293; please add a section on this.

Thanks for this comment. We added a short para on dormancy. It appeared, that this aspect - although very interesting - is not well studied, yet.

Round 2

Reviewer 1 Report

The manuscript has been improved.

Reviewer 2 Report

Authors have addresseed all my comments. I have no further comments.